
Materials Science

# Brushed creation of liquid marbles

Eric Shen Lin[1], Zhixiong Song[1], Jian Wern Ong[1], Hassan Ali Abid[1],
Oi Wah Liew[2] and Tuck Wah Ng[1]

[1] Monash University, Clayton, Australia
[2] Cardiovascular Research Institute, Singapore, Singapore

## ABSTRACT

A method where particulates are transferred *via* a cosmetic brush onto liquid drops created on a highly non-wetting substrate with a hole to generate talc and graphite liquid marbles (LMs) and talc-graphite Janus liquid marbles is described. van der Waals forces facilitated the attachment of particulates to the dry brush bristles. Subsequently, the surface tension forces that developed from particle interaction with water (which were $O(10^2)$ higher than the van der Waals forces) could then engender transfer of the particulates to the liquid-gas interface of the drop. Forces below 1 mN applied by a dangling foil on the LM ensured preservation of the drop shape when the force was removed. During the application of this force, the contact angles at the contact lines behaved differently from sessile drops that are inclined on surfaces. This preparation method portends the ability to automate the creation of LMs and Janus LMs for various applications.

# INTRODUCTION

A liquid marble (LM), in essence, comprises a liquid core liquid surrounded by a particle shell. Since the initial demonstration of liquid marbles encapsulated with hydrophobized Lycopodium and silica particles (*Aussillous & Quéré, 2001*), various other particles have been used as the shell component (*Terhemen, Augusta & Ezekwuaku, 2021*; *Alp, Alp & Aydogan, 2020*; *Li et al., 2017*; *Mahmoudi & Azizian, 2021*; *Dandan & Erbil, 2009*; *Yusa et al., 2014*; *Zhao et al., 2017*; *Wang et al., 2021*), imbuing the marble with wide-ranging functionalities in terms of wetting properties, electric charge, aggregate size and color. These advances have been highlighted in relation to their application and physical characteristics in various reviews (*Bormashenko, 2017*; *McHale & Newton, 2015*; *Bormashenko et al., 2013*).

The original method of LM creation involves dispensing a liquid droplet on a bed of powder and rolling it about (*Aussillous & Quéré, 2001*). This has remained the mainstay approach for generating LMs since it can be done with minimal instrumentation. Harnessing mechanical forces through vortex mixing to agitate the particle bed has been reported to improve the repeatability of LM formation (*Singh et al., 2021*). Apart from this, agitation using electrostatic forces have also been explored (*Liyanaarachchi et al., 2013*; *Lobel et al., 2020*). Janus liquid marbles (JLM) can be made to possess multi-response characteristics, such as being transportable on solid substrates and liquid water surfaces by

Corresponding author
Tuck Wah Ng, engngtw@gmail.com

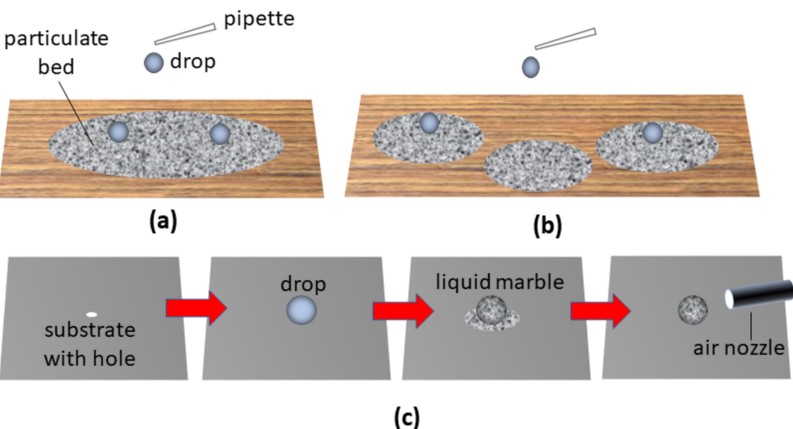

**Figure 1 Description of some of the current methods used to create liquid marbles.** In the classical method for LM formation, (A) drops of liquid are dispensed on the particle bed and moved around. The dispensation of drops onto separate particle beds (B) helps to limit transfer of material from one liquid body to another. A method that eschews the use of multiple particle beds (C) involves dispensing a drop on a non-wetting substrate with hole, depositing particles on the drop followed by removal of excess particles by blowing or by suction using an air nozzle.

magnetic forces, as well as the propensity for rupture upon exposure to either infrared radiation or acidic/basic vapors (*Xu et al., 2014*). They can be created using a two-step approach of controlled impact of liquid drops onto one particle bed and then onto another particle bed (*Lekshmi et al., 2020*) or the merging of two separate LMs (*Bormashenko et al., 2011*). These processes can however be highly stochastic.

High throughput generation of LMs can be challenging. Discharging multiple liquid drops onto a single particle bed (Fig. 1A) followed by agitation to achieve shell formation can result in material transference from one liquid body to another. This can be undesirable especially when the LMs are used as microbioreactors for the culture of cells, bacteria or viruses. Coalescence between liquid drops may be overcome somewhat by depositing the drops onto separate particle beds (Fig. 1B) followed by agitation instead, a strategy that can result in significant wastage since only a small fraction of particles will typically be taken up to coat the LM. In view of mounting evidence of the negative impact of airborne particle matter on health (*Liao et al., 2021*; *Peng et al., 2022*), there are clear benefits of reducing particle usage during LM generation to minimize their release to the environment.

An alternative approach for LM generation involves placement of a liquid drop on a highly non-wetting substrate with a hole, followed by deposition of powders over the drop and removing excess particles by air dislodgement or vacuum aspiration using an air nozzle (Fig. 1C) (*Lin et al., 2020*, *2021*). Although this technique prevents inter-drop material transfer and reduces particle wastage to some extent, uniformity of particle shell formation and potential dislodgement of the LM from a superhydrophobic surface remains a challenge.

Brushes have been applied since ancient Rome (*Corbishley, 2003*) to transfer cosmetic powder from a containment bed to the skin. With the appropriate brush material and application technique, it is possible to ensure highly uniform and optimal distributions of

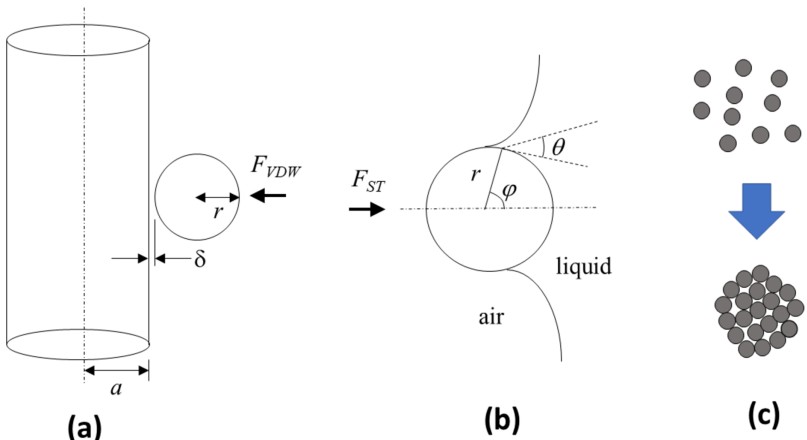

**Figure 2 Schematic depictions of theoretical models used to describe the van der Waals and surface tension forces that are key for the technique.** (A) van der Waals forces $F_{VDW}$ that foster particle adhesion to the brush bristles can be modelled as a sphere of radius $r$ separated by a distance $\delta$ from the surface of an infinitely long cylinder with radius $a$. (B) Surface tension forces $F_{ST}$ that facilitate particle adhesion to the liquid-gas interface is governed by the filling angle $\phi$ in relation to the contact angle $\theta$ and sphere radius $r$. (C) The particles, prior to participating in the physical processes to create a liquid marble, may exist as singular units or as agglomerated spheres.

particles over the skin surface. Brush feeders, operating in the same manner on industrial scales, have been applied for the transfer and dispersal of light, fluffy or low sphericity particles (*Barati, Jamshidi & Ebrahim, 2015*).

In this work, van der Waals (adhesion of solid particles to solid surfaces) and surface tension forces (attraction of solid particles to liquid-gas interfaces) are theoretically analyzed in the context of creating LMs and JLMs. The ability to create LMs and JLMs by directly transferring particles *via* a cosmetic brush onto liquid drops deposited on a highly non-wetting substrate with holes is then experimentally investigated. Experiments were also conducted to simulate and estimate the effect of the applied "brush-on" forces on the stability of the stationary liquid drop.

## THEORY

### Adhesion of a spherical particle to a cylindrical rod

When micron sized solid particles interact with the dry solid bristles of a brush, the van der Waals forces ($F_{VDW}$) play a primary role in engendering their mutual adhesion (*Hamaker, 1937*). An appropriate model to apply is that of a spherical particle of radius $r$ interacting with an infinitely long cylindrical rod of radius $a$ (see Fig. 2A). Suppose that the number of particles per volume unit is $n_c$ and $n_s$ for the cylinder and sphere, respectively. The interaction potential between the cylinder with that of the sphere can be expressed as

$$\varnothing = n_s \int_{V_s} \frac{\alpha}{\rho^6} dV_s \qquad (1)$$

where $\alpha$ = London-van der Waals constant, $\rho$ = distance of particles between two interacting bodies, $V_s$ = volume of the sphere, $V_c$ = volume of the cylinder. The energy resulting from the interacting potential is given by

$$E = n_c n_s \alpha \int_{V_s} \int_{V_c} \frac{dV_s dV_c}{\rho^6} = n_c \int_{V_c} \varnothing(r) dV_c \tag{2}$$

from which the interaction force can be determined using (*Kirsh, 2000*)

$$F_{vdw} = \frac{2AR^3}{3ax^2(x + 2R)^2} \tag{3}$$

where $A$ = Hamaker constant, $\delta$ = separation distance between sphere and cylinder, $R = r/a$, x = $\delta/a$. Based on Lifshitz's derivation, the Hamaker constant for the interaction of mediums 1 and 2 across medium 3 can be approximated using (*Leite et al., 2012*)

$$A \cong \frac{3}{4} k_B T \frac{(\varepsilon_1 - \varepsilon_3)(\varepsilon_2 - \varepsilon_3)}{(\varepsilon_1 + \varepsilon_3)(\varepsilon_2 + \varepsilon_3)} + \frac{3h\nu_e(n_1^2 - n_3^2)(n_2^2 - n_3^2)}{8\sqrt{2}\sqrt{n_1^2 + n_3^2}\sqrt{n_2^2 + n_3^2}\left[\sqrt{n_1^2 + n_3^2} + \sqrt{n_2^2 + n_3^2}\right]} \tag{4}$$

where $k_B$ = Boltzmann's constant ($1.380649 \times 10^{-23}$ m²kgs⁻²K⁻¹), $T$ = temperature in Kelvins, $h$ = Planck's constant ($6.62607015 \times 10^{-34}$ m²kgs⁻¹), $\nu_e$ = mean frequency, $\varepsilon_i$ = dielectric constant for medium $i$, and $n_i$ = refractive index of medium $i$.

### Adhesion of solid particles to liquid-gas interface

For LM creation, it is necessary for the solid particle to transfer from brush bristle to the liquid-gas interface of the liquid drop. The surface tension force ($F_{ST}$) plays a primary role in this (Fig. 2B). For a solid sphere of radius $r$ interacting with a liquid-gas interface with surface tension γ, the surface tension force is given by

$$F_{ST} = 2\pi\gamma r \sin\varphi \sin(\theta + \varphi) \tag{5}$$

where $\varphi$ = the fill angle and $\theta$ = contact angle. This assumption is based on inertia effects not playing a significant role. To verify this, consideration of the Bond number (*Bo*) is needed

$$Bo = \frac{\Delta\rho g L^2}{\gamma} \tag{6}$$

where $\Delta\rho$ = difference in density of the two phases, $g$ = gravitational acceleration, and $L$ = characteristic length.

## MATERIALS AND METHODS

### Substrate preparation

Candle soot was collected by placing an inverted beaker over a burning candle for a duration of 12 min (around 900 mg of soot collected during the process). A total of 15 ml of absolute ethanol was then poured into the beaker and thoroughly mixed using an ultrasonic cleaner (PS-20A; SUNNEX, Charlotte, NC, USA) for 5 min to synthesize the soot solution. Transparent polyvinyl chloride (PVC) slides (75 mm × 25 mm) of 0.5 mm thickness with holes of 2 mm diameter punched through were roughened (service provided by Dextech Technologies Pty Ltd, Noida, Uttar Pradesh) to $R_a$ values ranging

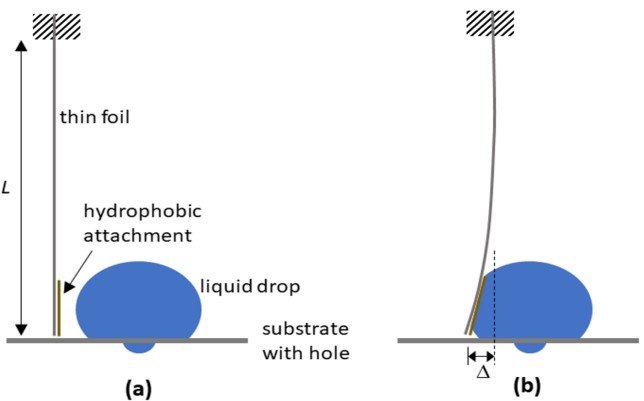

**Figure 3 Schematic depictions of how a dangling foil is able to determine the interaction forces with the drop.** The experiment to simulate the effect of brushing on forces on drop stability (A) involved using a thin foil of length $L$ dangled with its top end clamped firmly in position, and its bottom free end having a hydrophobic attachment. When the liquid drop is moved towards the foil and pushes against it (B), the free end of the foil undergoes lateral displacement $\Delta$, allowing the interaction force to be estimated.

from 0.1 to 0.2 µm. The slides were then sprayed with the soot solution using a compressed air driven air brush (Paasche, VL).

## Experimentation on LM and JLM creation

Drops of 50 µL volume deionized water were dispensed above the holes on the substrate. Cosmetic applicator brushes (Oxx Cosmetics) were then used to transfer hydrous magnesium silicate (Talc) powder (86255; Sigma, St. Louis, MO, USA; average size 45 µm) as well as graphite powder (282863; Sigma, St. Louis, MO, USA; average size 20 µm) to the drops to create LMs and JLMs. The bristles of the cosmetic applicator brush were made from polyester and had an average radius of 500 ± 23 µm, measured from 10 fiber samples using an optical microscope (BH2; Olympus, Tokyo, Japan).

## Experimentation on the simulated brush on force effect on liquid drop stability

Drops of 50 µL volume deionized water were dispensed above the holes on the substrate. A rectangular thin foil of length $L$ = 62 mm, width $w$ = 12 mm, and thickness $t$ = 0.5 mm created from material with modulus of elasticity $E$ = 70 × 10⁹ Pa, is dangled vertically with its top end clamped firmly in position (see Fig. 3A). A thin hydrophobic piece (using the same material prepared for the substrate) is attached to the free end of the foil. When the substrate with drop is moved towards the foil, the latter will deform elastically relative to its fixed top. The force $F_f$ imposed by the foil on the drop can be estimated from the deflection of the free end $\Delta$ using (*Ng & Panduputra, 2012*)

$$F_f = \frac{3EI\Delta}{L} \tag{7}$$

wherein $I$ the second moment of area is given by

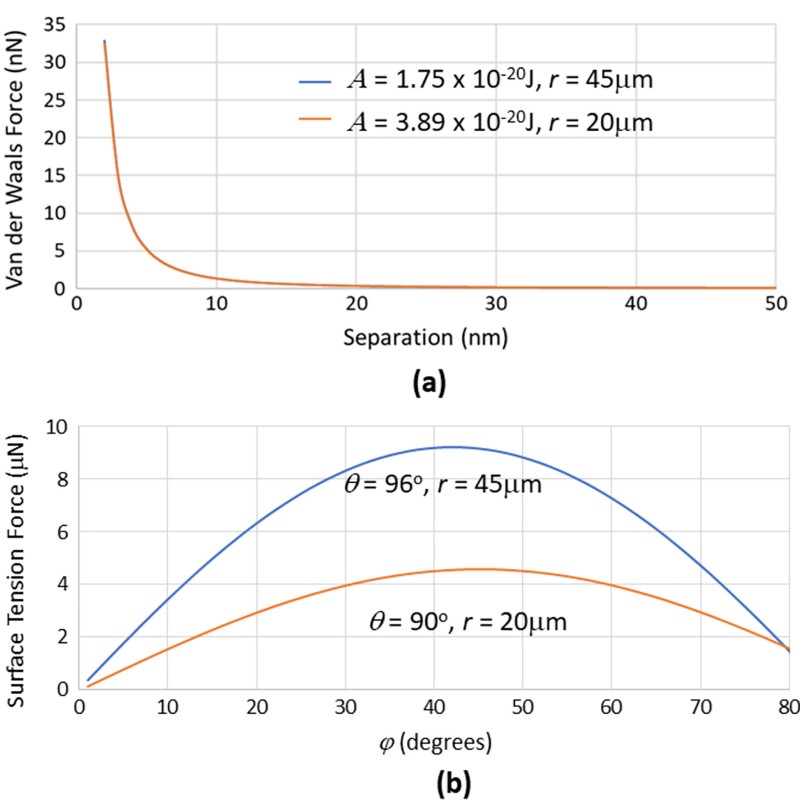

**Figure 4 Graphs of van der Waals forces (for dry powder attaching to the brush) and surface tension forces (for powder transfer to the drop) related to key parameters.** (A) van der Waals forces generated between a cylinder with radius = 500 μm interacting with spheres having different radius $r$ and Hamaker constants at various separation distances (between sphere and cylinder). These forces were much lower than (B) surface tension forces that are generated with spheres of different radius and contact angles at various filling angles.

$$I = \frac{wt^3}{12} \tag{8}$$

The contact angles, advancing and receding, exhibited by the drop were monitored by recording a time sequence of images with a camera (Moticam 2) as the thin foil interacted with the drop. The contact angles were measured using an imaging analysis software (ImageJ). The contact angles and drop shapes after specific levels of force interaction were also similarly monitored.

## RESULTS AND DISCUSSION

The calculations to estimate the van der Waals forces using Eq. (3) associated with particle pick up with the brush required knowledge of the Hamaker constants. Polyester (material of the brush fiber) and air were taken to be media 1 and 3 respectively, giving $\varepsilon_1 = 2.9$, $\varepsilon_3 = 1.006$, $n_1 = 1.56$, $n_3 = 1$. Based on a mean frequency $v_e = 6 \times 10^{14}$ Hz and temperature $T = 298$ K, having talc as the particle, in which $\varepsilon_2 = 1.8$ and $n_2 = 1.56$, gives $A = 1.75 \times 10^{-20}$ J using Eq. (4). Having graphite alternatively as the particle, in which $\varepsilon_2 = 12$ and $n_2 = 2.7$, gives $A = 3.89 \times 10^{-20}$ J using Eq. (4). As illustrated in Fig. 4A, van der Waals forces were

found to fall off steeply with increase in the separation distance (even at the nanometer range) between the sphere and cylinder, with virtually no difference between the talc-polyester and graphite-polyester cases. The calculations for surface tension forces using Eq. (4) alternatively were based on the use of contact angles of 96° and 90° for talc (*Rotenberg, Patel & Chandler, 2011*) and graphite (*Taherian et al., 2013*) respectively, determined previously by experimentation. As shown in Fig. 4B, the surface tension forces were significantly higher with talc than with graphite, although the contact angles for both materials were similar. This indicated the pronounced contribution of sphere radius in determining the force magnitude. It is noteworthy that the surface tension forces (in the micro-Newton range) were much higher than the van der Waals forces (which were in the nano-Newton range), which would indicate a favorable propensity of particles to be readily transferred from the bristles to the gas-liquid interface of the drop.

At this juncture, it is vital to note that the talc or graphite particles typically present themselves as singular units as well as agglomerates in a distribution. Agglomeration can result from van der Waals forces and capillary forces (due to moisture present in the environment) acting between the individual spheres to cause them to be physically bonded to each other to form granules. Ball mills are typically utilized in the manufacture of powders and they offer some measure of de-agglomeration to limit the formation of these granules in the process (*Blanc et al., 2020*). Nevertheless, granules may still be present in the product albeit would generally still be spherical in shape due to the rotating drum movement of the mill (*Vo et al., 2018*) (see Fig. 2C). If this were the case, the higher overall radius of the spherical granule (over the spherical particle) would impute lower van der Waals (Eq. (3)) and higher surface tension (Eq. (5)) forces that would further facilitate its transfer from the brush bristle to the liquid drop.

It is also apt to mention that the interaction of millimeter sized superhydrophobic spheres that are dropped onto the liquid-gas interface had been previously studied to interrogate their floating and sinking characteristics (*Lee & Kim, 2008*). The analysis in that work incorporated force contributions from the particle's weight, buoyancy and form drag, apart from surface tension. Since the particles here were not delivered in a manner where accelerations would be significant and their sizes were relatively miniscule (even for those that were agglomerated), the assumption that only surface tension forces being dominant is reasonable. The method of applying the particles here also permitted one side of the drop to be encapsulated and opaque whilst another side remaining transparent. That there were no particles observed to have breached the liquid-gas interface to enter into the liquid medium during the particle transfer process from the brush bristles in this way affirmed the claim that only the surface tension forces were significant. Finally, it is also pertinent that in using Eq. (6) based on the interaction of talc particle and water, the Bond numbers were found to be $O(10^{-4})$ and $O(10^{-2})$ with single and agglomerated talc particles (10× in size) respectively. These small values discount the contribution of inertia when the particles are transferred from the brush bristles to the liquid-gas interface.

The premise of the mechanics outlined is illustrated in the sequence of images furnished in Fig. 5. It is evident that brush bristles in its dry state is able to collect a sizeable amount of

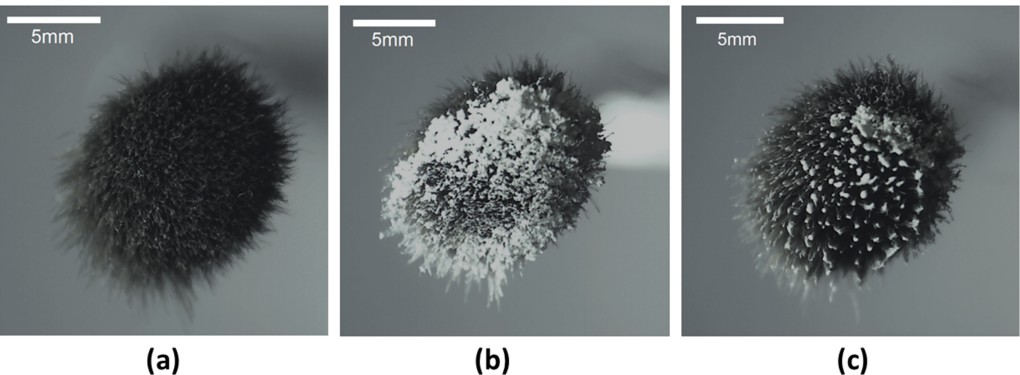

**Figure 5 Images of powder distribution on the brush using the technique.** Images of the cosmetic brush used in (A) its parent state, (B) following contact with a talc powder bed, and (C) after transference to the liquid-gas interface of a drop by contact.

talc particles from the powdered bed, and most of these particles can then be efficiently transferred to the liquid-gas interface of a drop.

The images presented in Fig. 6 clearly indicate the ability to create LMs and JLMs using the proposed technique. The distribution of particles on the liquid-gas interface were more uniform, the process more readily conducted, and less particles were wasted compared to previously reported methods. In addition, it was not only monolayer but spherical agglomerated granules (as described earlier) that could also be found (see inset of Fig. 6B). Due to the seemingly stable location of the drop over a hole on the substrate, it is conceivable for an array of drops to be created on the substrate followed by the brushing on of particles. This suggests a possibility of automating the process to increase LM and JLM production throughput.

A potential challenge in automating this "brush-on" process lies in the possibility of drop deformation and dislodgement when the interaction forces are applied. Experiments using a dangling thin foil provided a quantitative measure of the force magnitude applied. It was found that regardless of the stiffness of the foil, it was not possible to "sweep" the drop away as a single body. Instead, a portion of the liquid drop would be deformed and squeezed under the foil, allowing for a persistent contact line to be created there (see Fig. 7A). This behavior is attributed to a force being applied to one interface of the liquid body (from its left). This is distinct from the case of a sessile drop placed on a substrate that was then tilted, where the gravitational forces would be acting on the whole liquid body. In addition, the liquid used (in this case water) has low shear stress resistance which further exacerbates this phenomenon. More viscous or non-Newtonian fluids may permit the ability of displacement as a single body.

An assessment of drop stability in relation to the application of brush-on forces is needed to determine any shape change after the interaction force is removed (see Figs. 7B and 7C). From a series of experiments conducted, it was found that forces under 1 mN caused no change to the original contact angles of the drop. Hence, this constitutes the upper limit of applied particle brush-on forces to create LMs and JLMs.

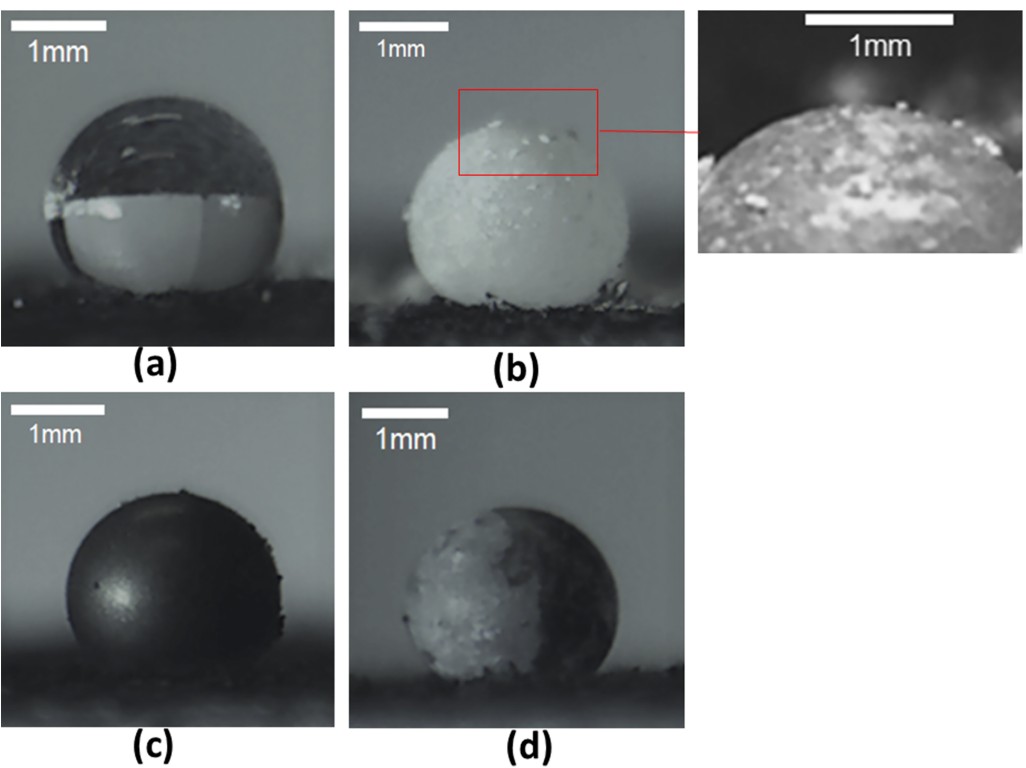

**Figure 6 Images of drop as well as liquid marbles and Janus liquid marbles created with the technique.** Images recorded of (A) a liquid drop. Using the proposed brush on method on the drop, liquid marbles were created with (B) talc and (C) graphite particles, as well as (D) Janus liquid marbles were created with talc and graphite particle encapsulation at the left and right sides respectively.

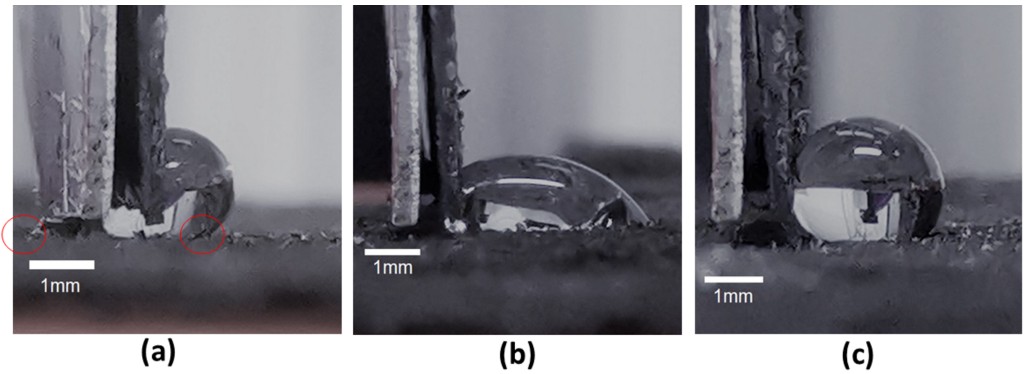

**Figure 7 Stages of interaction between drop and the dangling foil.** When interacted with the dangling thin foil, part of the body of the drop is always squeezed under the foil (A) such that the positions of the contact lines on the left and right are indicated with the red circles as shown. When the applied interaction force is relatively large and then released (B), the drop undergoes a departure from its initial shape and contact angles. Such a departure is eschewed when the applied interaction force is limited and then released (C).

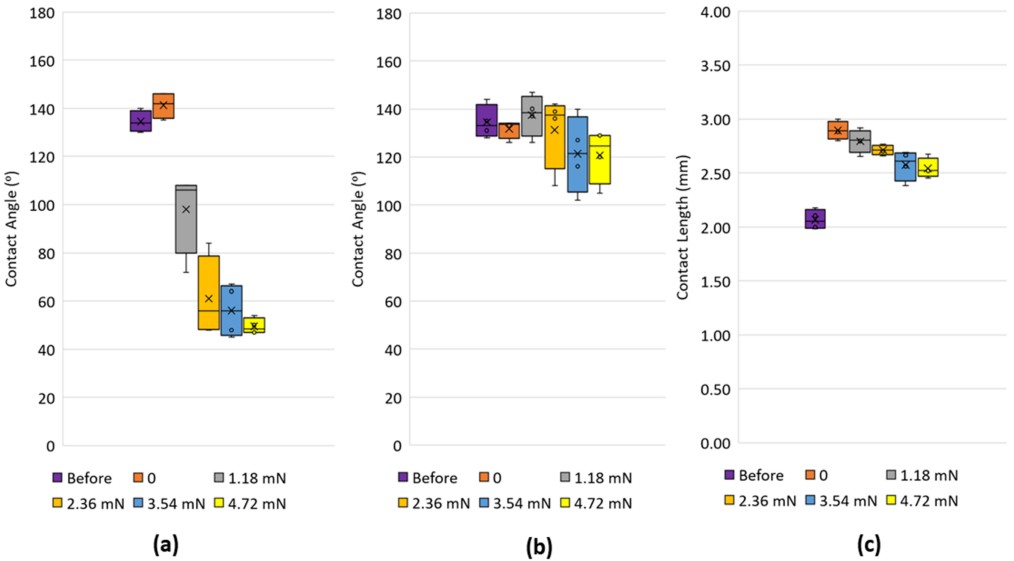

**Figure 8 Contact angles on the left and right of a drop interacting with the dangling foil.** Box plots of the contact angles measured at the left (A) and right (B) contact lines as well as contact length (C) before the foil contacts the drop, just when the foil contacts the drop, and when the foil imparts different levels of forces on the drop. Each plot is based on five separate readings.

The process of moving the dangling foil to contact with the drop and the subsequent increase in applied force on the drop produced unique contact angle trends at the left and right contact lines as depicted in Fig. 7A. Prior to contact, the drop exhibited contact angles averaging 135°. Upon immediate contact of the foil with the drop (force = 0), the left contact line exhibited slightly larger average contact angles (142°) compared with the right contact line (131°) (see Figs. 8A and 8B respectively). This seemingly counterintuitive outcome is due to the capillary flow of liquid squeezing into the gap between the foil and substrate, causing the left and right contact angles to be advancing and receding respectively. Upon the increase in force application through the foil, the contact angle at the left contact line reduced appreciably, reversing its role from being advancing to receding. Consequently, the contact angle at the right contact line now switches from receding to advancing. Cursorily, this appears to adhere to the mechanics of a sessile drop that is being progressively tilted. It should be noted, however, that the contact angle of the right contact line is reduced as the applied force increases, although more marginally, from the equilibrium contact angle (see Fig. 8B). With drops on inclined surfaces, the advancing contact angle is always greater than the equilibrium contact angle. This behavior indicates a mechanism in which the bottom part of the liquid body (demarcated by a virtual horizontal line that runs along the lowest part of the foil) was trying to accommodate to the force perturbation differently from its upper part. The former was essentially trying to "spread" itself to take advantage of its adhesion forces to the substrate, whereas the latter was attempting to adjust its shape in the absence of any adhesion forces. A similar demonstration of drops undergoing separate behaviors at its top and bottom was reported

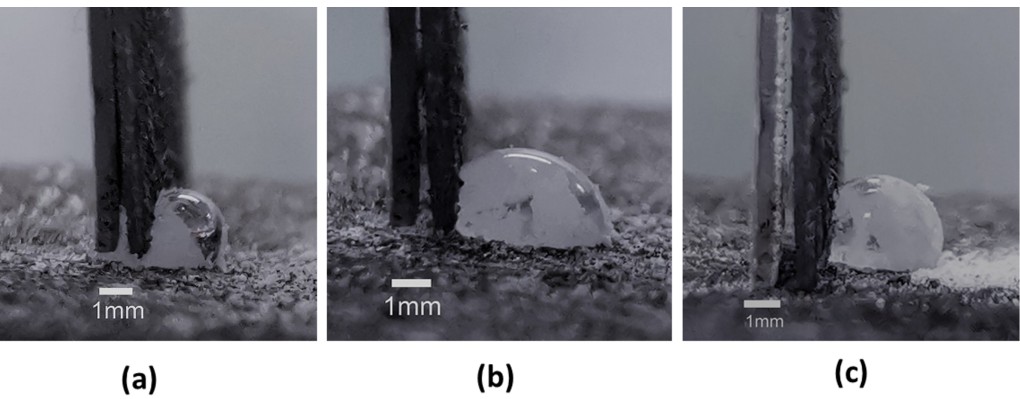

**(a)**          **(b)**          **(c)**

**Figure 9 Drop with particles interacting with the dangling foil.** For a drop that was partially encapsulated with particles, its interaction with the dangling thin foil was found to be similar to that of a drop without any particle encapsulation (see Fig. 7).

previously in another application context (*Katariya & Ng, 2013*; *Katariya, Vuong & Ng, 2014*). From tracking the contact length of the drop, it was found that the contact length of the drop increased appreciably as soon as contact with the foil was made. This is due to the capillary flow of liquid squeezing into the gap between the foil and substrate. Upon the increase in force application through the foil, this length reduced, although only marginally. This is attributed to some small depinning and re-pinning events occurring at the left contact line.

In a practical situation, it is not possible to encapsulate the drop with particles with a single stroke of the brush. Figure 9 presents the sequence of images for a drop that was initially partially encapsulated with talc particles and then contacted with the dangling foil. The similarities in the drop morphology with that shown in Fig. 7 imputes no change in drop characteristics are expected even as the particles were progressively being brushed on.

While the method is demonstrated here with talc and graphite, the use of other particle material such as PTFE, PVDF, or lycopodium is possible. In fact, the particles do not coat the liquid-gas interface as a single layer but rather as aggregates. Since this can trap air, it is possible to create LMs and JLMs that comprise moderately hydrophilic particles such as graphite through the manifestation of Cassie-like wetting mechanisms. While encapsulating both the overhead and overhanging components of the drop with particles is possible, one of the advantages in creating the LMs and JLMs on substrates with a hole with the overhanging component free of particle encapsulation is to allow for operations like electrolysis to be conducted there to supply on-demand aeration to the overhead component (*Lin et al., 2022*). Nevertheless, it is possible to encapsulate the overhanging component by brushing on particles there as well if needed. From experimentation conducted with different types of brushes, the volumetric density of the bristles did not exhibit any noticeable effect on LM and JLM creation.

It is important to note that liquid marbles with monolayer encapsulation are facilitated when the particles used are mono disperse and functionalized to be coalescence averse (*Asaumi et al., 2020*). In the experimentation conducted, the Talc particles were

transferred onto the liquid drop with the cosmetic brush in a manual fashion. Since the degree of granulation with the method here was lower than with particles dropped onto the drop followed by air dislodgement (*Lin et al., 2020, 2021*), it is conceivable that a robotic or machine-controlled delivery of the particles might not only ensure higher uniformity in particle coverage but also achieve higher process throughput as well. This will facilitate the creation of liquid marbles for various applications, particularly in those related to laboratory sensing where the marble does not need to be in motion (*Zhao et al., 2015; Arbatan et al., 2012*). It should also be noted that a previous work has shown that both rolling and compressing a drop on the particle bed helped to create more monolayered liquid marbles (*Shi & Li, 2020*). Due to the contact of the brush bristles on the drop resulting in some compressive deformation on the drop, it is conceivable that this may be a cause of the reduced particle granulation observed here.

## CONCLUSIONS

The ability to create talc and graphite liquid marbles and talc-graphite Janus liquid marbles directly by transferring particles *via* a cosmetic brush onto liquid drops created on highly non-wetting substrate with holes was demonstrated here. This method is physically supported by the van der Waals forces, that allowed particles to attach to the dry bristles of brushes, having magnitudes $O(10^2)$ lower than the surface tension forces that would typically allow their subsequent transfer onto the liquid-gas interface of the drop. Experiments conducted with a dangling foil cantilever showed that application forces below 1 mN would ensure preservation of the drop shape when the force was removed. Notwithstanding the stiffness of the foil, the drop could not be removed as a single body, with some part of it was always retained under the foil. On immediate contact of the foil with the drop, the contact angle at the contact line under the foil was advancing before switching to receding with greater application of force. These characteristics are different to those of a sessile drop that is inclined on a substrate.

## ACKNOWLEDGEMENTS

Specific resources used in this work have been generously provided by Dextech Technologies Pte Ltd.

### Funding

The authors received no funding for this work.

### Competing Interests

The authors declare that they have no competing interests.

### Author Contributions

- Eric Shen Lin conceived and designed the experiments, performed the experiments, analyzed the data, prepared figures and/or tables, authored or reviewed drafts of the article, and approved the final draft.

- Zhixiong Song performed the experiments, analyzed the data, prepared figures and/or tables, and approved the final draft.
- Jian Wern Ong performed the experiments, prepared figures and/or tables, and approved the final draft.
- Hassan Ali Abid performed the experiments, prepared figures and/or tables, and approved the final draft.
- Oi Wah Liew analyzed the data, authored or reviewed drafts of the article, and approved the final draft.
- Tuck Wah Ng conceived and designed the experiments, analyzed the data, prepared figures and/or tables, authored or reviewed drafts of the article, and approved the final draft.

## Data Availability

The raw data is available in a Supplemental File.

## Supplemental Information

Supplemental information for this article can be found online at http://dx.doi.org/10.7717/peerj-matsci.24#supplemental-information.

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
