# Peer review of "Brushed creation of liquid marbles"

_PeerJ Materials Science, doi:10.7717/peerj-matsci.24_

## Round 0.1 · original submission · Minor Revisions

Please revise the manuscript according the reviewers' comments.

Reviewer 1 ·

Basic reporting

In this manuscript, the authors used a cosmetic brush to transfer particulates to the liquid drops on a highly non-wetting substrate. Through reasonable theoretical calculation, this manuscript proves that the van der Waals force between the cosmetic brush and the particles is lower than the surface tension between the particles and the droplets, thus, the particles separate from the cosmetic brush and agglomerate on the surface of the droplets. The marble has wide-ranging functionalities in terms of wetting properties, electric charge, aggregate size, and color. I think this manuscript can be accepted after the following questions were addressed.
Different brushing particle sizes will have different effects on the adhesion of particles, so artificial brushing will certainly produce uneven results. Can the uniformity of particles be guaranteed by brushing with a cosmetic brush? Is the brushing process controlled by a machine or operated by manual work? If a machine was used, please introduce it in detail.

Experimental design

In this manuscript, the authors used a cosmetic brush to transfer particulates to the liquid drops on a highly non-wetting substrate. Through reasonable theoretical calculation, this manuscript proves that the van der Waals force between the cosmetic brush and the particles is lower than the surface tension between the particles and the droplets, thus, the particles separate from the cosmetic brush and agglomerate on the surface of the droplets. The marble has wide-ranging functionalities in terms of wetting properties, electric charge, aggregate size, and color. I think this manuscript can be accepted after the following questions were addressed.
Different brushing particle sizes will have different effects on the adhesion of particles, so artificial brushing will certainly produce uneven results. Can the uniformity of particles be guaranteed by brushing with a cosmetic brush? Is the brushing process controlled by a machine or operated by manual work? If a machine was used, please introduce it in detail.

Validity of the findings

In this manuscript, the authors used a cosmetic brush to transfer particulates to the liquid drops on a highly non-wetting substrate. Through reasonable theoretical calculation, this manuscript proves that the van der Waals force between the cosmetic brush and the particles is lower than the surface tension between the particles and the droplets, thus, the particles separate from the cosmetic brush and agglomerate on the surface of the droplets. The marble has wide-ranging functionalities in terms of wetting properties, electric charge, aggregate size, and color. I think this manuscript can be accepted after the following questions were addressed.
Different brushing particle sizes will have different effects on the adhesion of particles, so artificial brushing will certainly produce uneven results. Can the uniformity of particles be guaranteed by brushing with a cosmetic brush? Is the brushing process controlled by a machine or operated by manual work? If a machine was used, please introduce it in detail.

Reviewer 2 ·

Basic reporting

The manuscript is entitled “Brushed creation of liquid marbles”. In this work, the authors report the preparation of talc and graphite liquid marbles (LMs) and talc-graphite Janus liquid marbles This research is follow on work of several publications of this type in the last few years.

Experimental design

Your theory and experimental section needs more detail.

Validity of the findings

Generally speaking, the paper is generally well-written and the experimental work is mostly sound. I feel that this manuscript is suitable for publication.

Additional comments

The authors are encouraged to give some comments on the application and future development of the liquid marbles in the Introduction section. Some related references can be referred. Such as, Yan Zhao et al., Advanced Functional Material, 2015, 25, 437; Junfei Tian et al. Chemical Communication, 2011, 46, 4734; Advanced Healthcare Materials, 2012, 1, 80.

Reviewer 3 ·

Basic reporting

This paper describes a new method for liquid marble preparation, which is based on a simpe brushing process. The study is comprehensive while the following comments need to be addressed.
1. There are many unprecise expressions. eg. " the surface tension forces arising from particle interaction with water engendered transfer"
2. When discussing the brush pushing process related to Fig. 3 or 7, a similar manipulation reported previously could be a useful reference. See Advanced Materials Interfaces, 2020, 7, 2001081
3. The figures in this paper are not well produced.
4. Figure 5 does not sufficiently match the discussion in the text. Sequenced images should also include the evolution of droplets instead of just showing the brush images.

Experimental design

good

Validity of the findings

good

---

## Round 0.2 · Minor Revisions

Some minor revisions are still needed.

Reviewer 1 ·

Basic reporting

well noted

Experimental design

well noted

Validity of the findings

well noted

Additional comments

well noted

Reviewer 2 ·

Basic reporting

The authors have revised the manuscript. It can be accepted.

Experimental design

Good

Validity of the findings

Good

Reviewer 3 ·

Basic reporting

I don't think the authors have well addressed the reviewers' comments. Some suggestions are not taken but the reasons are not convictive. For example, they argue that the method in Advanced Materials Interfaces, 2020, 7, 2001081 is different from their method. Of course the two methods are generally different, but the pushing process related to Figure 8 in the recommended literature could be discussed by comparing with their pushing process.

Experimental design

fair

Validity of the findings

fair

---

## Round 0.3 · accepted · Accept

The manuscript is acceptable in the current version.